# Neurosymbolic Transformers for Multi-Agent Communication

**Jeevana Priya Inala** [1][*]    **Yichen Yang** [1][*]    **James Paulos** [2]    **Yewen Pu** [1]

**Osbert Bastani** [2]    **Vijay Kumar** [2]    **Martin Rinard** [1]    **Armando Solar-Lezama** [1]

[1] MIT CSAIL          [2] University of Pennsylvania

{jinala,yicheny,yewenpu,rinard,asolar}@csail.mit.edu
{jpaulos, obastani, kumar}@seas.upenn.edu

## Abstract

We study the problem of inferring communication structures that can solve cooperative multi-agent planning problems while minimizing the amount of communication. We quantify the amount of communication as the maximum degree of the communication graph; this metric captures settings where agents have limited bandwidth. Minimizing communication is challenging due to the combinatorial nature of both the decision space and the objective; for instance, we cannot solve this problem by training neural networks using gradient descent. We propose a novel algorithm that synthesizes a control policy that combines a programmatic communication policy used to generate the communication graph with a transformer policy network used to choose actions. Our algorithm first trains the transformer policy, which implicitly generates a "soft" communication graph; then, it synthesizes a programmatic communication policy that "hardens" this graph, forming a *neurosymbolic transformer*. Our experiments demonstrate how our approach can synthesize policies that generate low-degree communication graphs while maintaining near-optimal performance.

## 1 Introduction

Many real-world robotics systems are distributed, with teams of agents needing to coordinate to share information and solve problems. Reinforcement learning has recently been demonstrated as a promising approach to automatically solve such multi-agent planning problems [28, 16, 8, 18, 9, 13].

A key challenge in (cooperative) multi-agent planning is how to coordinate with other agents, both deciding whom to communicate with and what information to share. One approach is to let agents communicate with all other agents; however, letting agents communicate arbitrarily can lead to poor generalization [12, 21]; furthermore, it cannot account for physical constraints such as limited bandwidth. A second approach is to manually impose a communication graph on the agents, typically based on distance [12, 29, 21, 26]. However, this manual structure may not reflect the optimal communication structure—for instance, one agent may prefer to communicate with another one that is farther away but in its desired path. A third approach is to use a transformer [31] as the policy network [4], which uses attention to choose which other agents to focus on. However, since the attention is soft, each agent still communicates with every other agent.

---

[*]Equal contribution.

We study the problem of learning a communication policy that solves a multi-agent planning task while minimizing the amount of communication required. We measure the amount of communication on a given step as the maximum degree (in both directions) of the communication graph on that step; this metric captures the maximum amount of communication any single agent must perform at that step. While we focus on this metric, our approach easily extends to handling other metrics—e.g., the total number of edges in the communication graph, the maximum in-degree, and the maximum out-degree, as well as general combinations of these metrics.

A key question is how to represent the communication policy; in particular, it must be sufficiently expressive to capture communication structures that both achieve high reward and has low communication degree, while simultaneously being easy to train. Neural network policies can likely capture good communication structures, but they are hard to train since the maximum degree of the communication graph is a discrete objective that cannot be optimized using gradient descent. An alternative is to use a structured model such as a decision tree [3] or rule list [34] and train using combinatorial optimization. However, these models perform poorly since choosing whom to communicate with requires reasoning over sets of other agents—e.g., to avoid collisions, an agent must communicate with its nearest neighbor in its direction of travel.

We propose to use programs to represent communication policies. In contrast to rule lists, our programmatic polices include components such as filter and map that operate over sets of inputs. Furthermore, programmatic policies are discrete in nature, making them amenable to combinatorial optimization; in particular, we can compute a programmatic policy that minimizes the communication graph degree using a stochastic synthesis algorithm [25] based on MCMC sampling [19, 10].

A key aspect of our programs is that they can include a random choice operator. Intuitively, random choice is a key ingredient needed to minimize the communication graph degree without global coordination. For example, suppose there are two groups of agents, and each agent in group $A$ needs to communicate with an agent in group $B$, but the specific one does not matter. Using a deterministic communication policy, since the same policy is shared among all agents, each agent in group $A$ might choose to communicate with the same agent $j$ in group $B$ (e.g., if agents in the same group have similar states). Then, agent $j$ will have a very high degree in the communication graph, which is undesirable. In contrast, having each agent in group $A$ communicate with a uniformly random agent in group $B$ provides a near-optimal solution to this problem, without requiring the agents to explicitly coordinate their decisions.

While we can minimize the communication graph degree using stochastic search, we still need to choose actions based on the communicated information. Thus, we propose a learning algorithm that integrates our programmatic communication policy with a transformer policy for selecting actions. We refer to the combination of the transformer and the programmatic communication policy as a *neurosymbolic transformer*. This algorithm learns the two policies jointly. At a high level, our algorithm first trains a transformer policy for solving the multi-agent task; as described above, the soft attention weights capture the extent to which an edge in the communication graph is useful. Next, our algorithm trains a programmatic communication policy that optimizes both goals: (i) match the transformer as closely as possible, and (ii) minimize the maximum degree of the communication graph at each step. In contrast to the transformer policy, this communication policy makes hard decisions about which other agents to communicate with. Finally, our algorithm re-trains the weights of the transformer policy, except where the (hard) attention weights are chosen by the communication policy.

We evaluate our approach on several multi-agent planning tasks that require agents to coordinate to achieve their goals. We demonstrate that our algorithm learns communication policies that achieve task performance similar to the original transformer policy (i.e., where each agent communicates with every other agent), while significantly reducing the amount of communication. Our results demonstrate that our algorithm is a promising approach for training policies for multi-agent systems that additionally optimize combinatorial properties of the communication graph [2]

**Example.** Consider the example in Figure 1, where agents in group 1 (blue) are navigating from the left to their goal on the right, while agents in group 2 (red) are navigating from the right to their goal on the left. In this example, agents have noisy observations of the positions of other agents (e.g.,

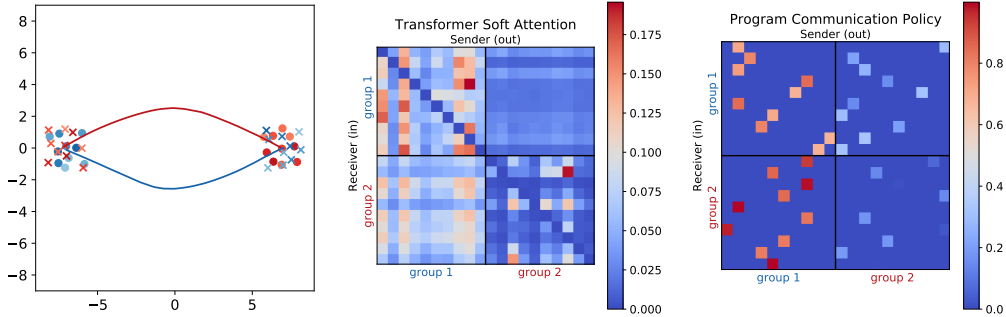

Figure 1: Left: Two groups of agents (red vs. blue) at their initial positions (circles) trying to reach their goal positions (crosses). Agents must communicate both within group and across groups to choose a collision free path to take (solid line shows a path for a single agent in each group). Middle: The soft attention weights of the transformer policy computed by the agent along the $y$-axis for the message received from the agent along the $x$-axis for the initial step. Right: The hard attentions learned by the programmatic communication policy to imitate the transformer.

based on cameras or LIDAR); however, they do not have access to internal information such as their planned trajectories or even their goals. Thus, to solve this task, each agent must communicate both with its closest neighbor in the same group (to avoid colliding with them), as well as with any agent in the opposite group (to coordinate so their trajectories do not collide). The communication graph of the transformer policy (in terms of soft attention weights) is shown in Figure 1 (middle); every agent needs to communicate with all other agents. The programmatic communication policy synthesized by our algorithm is

$$\operatorname{argmax}(\operatorname{map}(-d^{i,j}, \operatorname{filter}(\theta^{i,j} \geq -1.85, l))), \qquad \operatorname{random}(\operatorname{filter}(d^{i,j} \geq 3.41, l)).$$

Agent $i$ uses this program to choose two other agents $j$ from the list of agents $\ell$ from whom to request information; $d^{i,j}$ is the distance between them and $\theta^{i,j}$ is the angle between them. The first rule chooses the nearest agent $j$ (besides itself) such that $\theta^{i,j} \in [-1.85, \pi]$, and the second chooses a random agent in the other group. The communication graph is visualized in Figure 1 (right).

**Related work.** There has been a great deal of recent interest in using reinforcement learning to automatically infer good communication structures for solving multi-agent planning problems [12, 29, 21, 4, 26]. Much of this work focuses on inferring what to communicate rather than whom to communicate with; they handcraft the communication structure to be a graph (typically based on distance) [12, 29, 21], and then use a graph neural network [24, 14] as the policy network. There has been some prior work using transformer network to infer the communication graph [4]; however, they rely on soft attention, so the communication graph remains fully connected. Prior work [27] frames the multi-agent communication problem as a MDP problem where the decisions of when to communicate are part of the action space. However, in our case, we want to learn who to communicate with in-addition to when to communicate. This results in a large discrete action space and we found that RL algorithms perform poorly on this space. Our proposed approach addresses this challenge by using the transformer as a teacher.

There has also been a great deal of interest using an oracle (in our case, the transformer policy) to train a policy (in our case, the programmatic communication policy) [15, 20]. In the context of multi-agent planning, this approach has been used to train a decentralized control policy using a centralized one [29]; however, their communication structure is manually designed.

Finally, in the direction of program synthesis, there has been much recent interest in leveraging program synthesis in the context of machine learning [7, 6, 30, 22, 35, 17]. Specifically in the context of reinforcement learning, it has been used to synthesize programmatic control policies that are more interpretable [33, 32], that are easier to formally verify [2, 23], or that generalize better [11]; to the best of our knowledge, none of these approaches have considered multi-agent planning problems.

## 2 Problem Formulation

We formulate the multi-agent planning problem as a decentralized partially observable Markov decision process (POMDP). We consider $N$ agents $i \in [N] = \{1, ..., N\}$ with states $s^i \in \mathcal{S} \subseteq \mathbb{R}^{d_S}$, actions $a^i \in \mathcal{A} \subseteq \mathbb{R}^{d_A}$, and observations $o^{i,j} \in \mathcal{O} \subseteq \mathbb{R}^{d_O}$ for every pair of agents ($j \in [N]$). Following prior work [18, 9, 4], we operate under the premise of centralized training and decentralized execution. Hence, during training the POMDP has global states $\mathcal{S}^N$, global actions $\mathcal{A}^N$, global observations $\mathcal{O}^{N \times N}$, transition function $f : \mathcal{S}^N \times \mathcal{A}^N \to \mathcal{S}^N$, observation function $h : \mathcal{S}^N \to \mathcal{O}^{N \times N}$, initial state distribution $s_0 \sim \mathcal{P}_0$, and reward function $r : \mathcal{S}^N \times \mathcal{A}^N \to \mathbb{R}$.

The agents all use the same policy $\pi = (\pi^C, \pi^M, \pi^A)$ divided into a *communication policy* $\pi^C$ (choose other agents from whom to request information), a *message policy* $\pi^M$ (choose what messages to send to other agents), and an *action policy* $\pi^A$ (choose what action to take). Below, we describe how each agent $i \in [N]$ chooses its action $a^i$ at any time step.

**Step 1 (Choose communication).** The communication policy $\pi^C : \mathcal{S} \times \mathcal{O}^N \to \mathcal{C}^K$ inputs the state $s^i$ of current agent $i$ and its observations $o^i = (o^{i,1}, ..., o^{i,N})$, and outputs $K$ other agents $c^i = \pi^C(s^i, o^i) \in \mathcal{C}^K = [N]^K$ from whom to request information. The *communication graph* $c = (c^1, ..., c^N) \in \mathcal{C}^{N \times K}$ is the directed graph $G = (V, E)$ with nodes $V = [N]$ and edges $E = \{j \to i \mid (i, j) \in [N]^2 \wedge j \in \pi^C(s^i, o^i)\}$. For example, in the communication graph in Figure 1 (right), $c^0 = (1, 19)$—i.e., agent 0 in group 1 receives messages from agent 1 in group 1 and agent 19 in group 2.

**Step 2 (Choose and send/receive messages).** For every other agent $j \in [N]$, the message policy $\pi^M : \mathcal{S} \times \mathcal{O} \to \mathcal{M}$ inputs $s^i$ and $o^{i,j}$ and outputs a message $m^{i \to j} = \pi^M(s^i, o^{i,j})$ to be sent to $j$ if requested. Then, agent $i$ receives messages $m^i = \{m^{j \to i} \mid j \in c^i\} \in \mathcal{M}^K$.

**Step 3 (Choose action).** The action policy $\pi^A : \mathcal{S} \times \mathcal{O}^N \times \mathcal{M}^K \to \mathcal{A}$ inputs $s^i$, $o^i$, and $m^i$, and outputs action $a^i = \pi^A(s^i, o^i, m^i)$ to take.

**Sampling a trajectory/rollout.** Given initial state $s_0 \sim \mathcal{P}_0$ and time horizon $T$, $\pi$ generates the trajectory $(s_0, s_1, ..., s_T)$, where $o_t = h(s_t)$ and $s_{t+1} = f(s_t, a_t)$, and where for all $i \in [N]$, we have $c_t^i = \pi^C(s_t^i, o_t^i)$, $m_t^i = \{\pi^M(s_t^j, o_t^{j,i}) \mid j \in c_t^i\}$, and $a_t^i = \pi^A(s_t^i, o_t^i, m_t^i)$.

**Objective.** Then, our goal is to train a policy $\pi$ that maximizes the objective

$$J(\pi) = J^R(\pi) + \lambda J^C(\pi) = \mathbb{E}_{s_0 \sim \mathcal{P}_0}\left[\sum_{t=0}^T \gamma^t r(s_t, a_t)\right] - \lambda \mathbb{E}_{s_0 \sim \mathcal{P}_0}\left[\sum_{t=0}^T \max_{i \in [N]} \deg(i; c_t)\right],$$

where $\lambda \in \mathbb{R}_{>0}$ is a hyperparameter, the reward objective $J^R$ is the time-discounted expected cumulative reward over time horizon $T$ with discount factor $\gamma \in (0, 1)$, and the communication objective $J^C$ is to minimize the degree of the communication graph, where $c_t$ is the communication graph on step $t$, and $\deg(i; c_t)$ is the sum of the incoming and outgoing edges for node $i$ in $c_t$. Each agent computes its action based on just its state, its observations of other agents, and communications received from the other agents; thus, the policy can be executed in a decentralized way.

**Assumptions on the observations of other agents.** We assume that $o^{i,j}$ is available through visual observation (e.g., camera or LIDAR), and therefore does not require extra communication. In all experiments, we use $o^{i,j} = x^j - x^i + \epsilon^{i,j}$—i.e., the position $x^j$ of agent $j$ relative to the position $x^i$ of agent $i$, plus i.i.d. Gaussian noise $\epsilon^{i,j}$. This information can often be obtained from visual observations (e.g., using an object detector); $\epsilon^{i,j}$ represents noise in the visual localization process.

The observation $o^{i,j}$ is necessary since it forms the basis for agent $i$ to decide whether to communicate with agent $j$; if it is unavailable, then $i$ has no way to distinguish the other agents. If $o^{i,j}$ is unavailable for a subset of agents $j$ (e.g., they are outside of sensor range), we could use a mask to indicate that the data is missing. We could also replace it with alternative information such as the most recent message from $j$ or the most recent observation of $j$.

We emphasize that $o^{i,j}$ does not contain important internal information available to the other agents—e.g., their chosen goals and their planned actions/trajectories. This additional information is critical for the agents to coordinate their actions and the agents must learn to communicate such information.

# 3 Neurosymbolic Transformer Policies

Our algorithm has three steps. First, we use reinforcement learning to train an oracle policy based on transformers (Section 3.1), which uses soft attention to prioritize edges in the communication graph. However, attention is soft, so every agent communicates with every other agent. Second, we synthesize a programmatic communication policy (Section 3.2) by having it mimic the transformer (Section 3.4). Third, we combine the programmatic communication policy with the transformer policy by overriding the soft attention of the transformer by the hard attention imposed by the programmatic communication policy (Section 3.3), and re-train the transformer network to fine-tune its performance (Section 3.5). We discuss an extension to multi-round communications in Appendix A.

## 3.1 Oracle Policy

We begin by training an *oracle policy* that guides the synthesis of our programmatic communication policy. Our oracle is a neural network policy $\pi_\theta = (\pi_\theta^C, \pi_\theta^M, \pi_\theta^A)$ based on the transformer architecture [31]; its parameters $\theta$ are trained using reinforcement learning to optimize $J^R(\pi_\theta)$. At a high level, $\pi_\theta^C$ communicates with all other agents, $\pi_\theta^M$ is the value computed by the transformer for pairs $(i,j) \in [N]^2$, and $\pi^A$ is the output layer of the transformer. While each agent $i$ receives information from every other agent $j$, the transformer computes a soft attention score $\alpha^{j \to i}$ that indicates how much weight agent $i$ places on the message $m^{j \to i}$ from $j$.

More precisely, the communication policy is $c^i = \pi_\theta^C(s^i, o^i) = [N]$ and the message policy is $m^{i \to j} = \pi_\theta^M(s^i, o^{i,j})$. The action policy is itself composed of a *key network* $\pi_\theta^K : \mathcal{S} \times \mathcal{O} \to \mathbb{R}^d$, a *query network* $\pi_\theta^Q : \mathcal{S} \to \mathbb{R}^d$, and an *output network* $\pi_\theta^O : \mathcal{S} \times \mathcal{M} \to \mathcal{A}$; then, we have

$$\pi_\theta^A(s^i, o^i, m^i) = \pi_\theta^O\left(s^i, \ \sum_{j=1}^N \alpha^{j \to i} m^{j \to i}\right), \tag{1}$$

where the attention $\alpha^{j \to i} \in [0,1]$ of agent $i$ to the message $m^{j \to i}$ received from agent $j$ is

$$(\alpha^{1 \to i}, ..., \alpha^{N \to i}) = \text{softmax}\left(\frac{\langle q^i, k^{i,1}\rangle}{\sqrt{d}}, ..., \frac{\langle q^i, k^{i,N}\rangle}{\sqrt{d}}\right), \tag{2}$$

where $k^{i,j} = \pi_\theta^K(s^i, o^{i,j})$ and $q^i = \pi_\theta^Q(s^i)$ for all $i,j \in [N]$. Since $\pi_\theta$ is fully differentiable, we can use any reinforcement learning algorithm to train $\theta$; assuming $f$ and $r$ are known and differentiable, we use a model-based reinforcement learning algorithm that backpropagates through them [5, 1].

## 3.2 Programmatic Communication Policies

Due to the combinatorial nature of the communication choice and the communication objective, we are interested in training communication policies represented as programs. At a high level, our communication policy $\pi_P^C$ is a parameterized program $P : \mathcal{S} \times \mathcal{O}^N \to \mathcal{C}^K$, which is a set of $K$ rules $P = (R_1, ..., R_K)$ where each rule $R : \mathcal{S} \times \mathcal{O}^N \to \mathcal{C}$ selects a single other agent from whom to request information—i.e., $\pi_P^C(s^i, o^i) = P(s^i, o^i) = (R_1(s^i, o^i), ..., R_K(s^i, o^i))$.

When applied to agent $i$, each rule first constructs a list

$$\ell = (x_1, ..., x_N) = ((s^i, o^{i,1}, 1), ..., (s^i, o^{i,N}, N)) \in \mathcal{X}^N,$$

where $\mathcal{X} = \mathcal{S} \times \mathcal{O} \times [N]$ encodes an observation of another agent. Then, it applies a combination of standard list operations to $\ell$—in particular, we consider two combinations

$$R ::= \text{argmax}(\text{map}(F, \text{filter}(B, \ell))) \mid \text{random}(\text{filter}(B, \ell)).$$

Intuitively, the first kind of rule is a deterministic aggregation rule, which uses $F$ to score every agent after filtering and then chooses the one with the best score, and the second kind of rule is a nondeterministic choice rule which randomly chooses one of the other agents after filtering.

More precisely, filter outputs the list of elements $x \in \ell$ such that $B(x) = 1$, where $B : \mathcal{X} \to \{0, 1\}$. Similarly, map outputs the list of pairs $(F(x), j)$ for $x = (s^i, o^{i,j}, j) \in \ell$, where $F : \mathcal{X} \to \mathbb{R}$. Next, argmax inputs a list $((v^{j_1}, j_1), ..., (v^{j_H}, j_H)) \in (\mathbb{R} \times [N])^H$, where $v^j$ is a score computed for

agent $j$, and outputs the agent $j$ with the highest score $v^j$. Finally, random($\ell$) takes as input a list $((s^i, o^{i,j_1}, j_1), ..., (s^i, o^{i,j_H}, j_H)) \in \mathcal{X}^H$, and outputs $j_h$ for a uniformly random $h \in [H]$.

Finally, the filter predicates $B$ and map functions $F$ have the following form:

$$B ::= \langle \beta, \phi(s^i, o^{i,j}) \rangle \geq 0 \mid B \wedge B \mid B \vee B \qquad F ::= \langle \beta, \phi(s^i, o^{i,j}) \rangle,$$

where $\beta \in \mathbb{R}^{d'}$ are weights and $\phi : \mathcal{S} \times \mathcal{O} \to \mathbb{R}^{d'}$ is a feature map.

### 3.3 Combined Transformer & Programmatic Communication Policies

A programmatic communication policy $\pi_P^C$ only chooses which other agents to communicate with; thus, we must combine it with a message policy and an action policy. In particular, we combine $\pi_P^C$ with the transformer oracle $\pi_\theta$ to form a combined policy $\pi_{P,\theta} = (\pi_P^C, \pi_\theta^M, \pi_{P,\theta}^A)$, where (i) the oracle communication policy $\pi_\theta^C$ is replaced with our programmatic communication policy $\pi_P^C$, and (ii) we use $\pi_P^C$ to compute hard attention weights that replace the soft attention weights in $\pi_{P,\theta}^A$.

The first modification is straightforward; we describe the second in more detail. First, we use $\pi_P^C$ as a mask to get the hard attention weights:

$$\alpha^{P,j \to i} = \begin{cases} \alpha^{j \to i}/Z & \text{if } j \in \pi_P^C(s^i, o^i) \\ 0 & \text{otherwise} \end{cases} \qquad \text{where} \qquad Z = \sum_{j \in \pi_P^C(s^i, o^i)} \alpha^{j \to i}$$

Now, we can use $\alpha^{P,j \to i}$ in place of $\alpha^{j \to i}$ when computing the action using $\pi_\theta$—i.e.,

$$\pi_{P,\theta}^A(s^i, m^i) = \pi_\theta^O \left( s^i, \sum_{j=1}^{N} \alpha^{P,j \to i} m^{j \to i} \right),$$

where the messages $m^{j \to i} = \pi_\theta^M(s^j, o^{j,i})$ are as before, and $\pi_\theta^O$ is the output network of $\pi_\theta^A$.

### 3.4 Synthesis Algorithm

To optimize $P$ over the search space of programs, we use a combinatorial search algorithm based on MCMC [19, 10, 25]. Given our oracle policy $\pi_\theta$, our algorithm maximizes the surrogate objective

$$\tilde{J}(P; \theta) = \tilde{J}^R(P; \theta) + \tilde{\lambda}\tilde{J}^C(P) = -\mathbb{E}_{s_0 \sim \mathcal{P}_0}\left[\sum_{t=0}^{T} \|a_t - a_t^P\|_1\right] - \tilde{\lambda}\mathbb{E}_{s_0 \sim \mathcal{P}_0}\left[\sum_{t=0}^{T} \max_{i \in [N]} \deg(i; c_t^P)\right],$$

where $\tilde{\lambda} \in \mathbb{R}_{>0}$ is a hyperparameter, the surrogate reward objective $\tilde{J}^R$ aims to have the actions $a_t^P$ output by $\pi_{P,\theta}$ match the actions $a_t$ output by the $\pi_\theta$, and the surrogate communication objective $\tilde{J}^C$ aims to minimize the degree of the communication graph $c_t^P$ computed using $\pi_P^C$.

Finally, a key to ensuring MCMC performs well is for sampling candidate programs $P$ and evaluating the objective $\tilde{J}(P)$ to be very efficient. The former is straightforward; our algorithm uses standard choice of neighbor programs that can be sampled very efficiently. For the latter, we precompute a dataset of tuples $D = \{(s, o, \alpha, a)\}$ by sampling trajectories of horizon $T$ using the oracle policy $\pi_\theta$. Given a tuple in $D$ and a candidate program $P$, we can easily compute the corresponding values $(a^P, \alpha^P, c^P)$. Thus, we can evaluate $\tilde{J}(P)$ using these values.

Note that, we sample trajectories using the transformer policy rather than using a program policy. The latter approach is less efficient because we have to sample trajectories at every iteration of the MCMC algorithm and we cannot batch process the objective metric across timesteps. The only potential drawback of sampling using the transformer policy is that the objective $\tilde{J}(P)$ can be affected by the shift in the trajectories. However, as our experiments demonstrate, we achieve good results despite any possible distribution shift; hence, the efficiency gains far outweigh any cons.

### 3.5 Re-training the Transformer

Once our algorithm has synthesized a program $P$, we can form the combined policy $\pi_{P,\theta}$ to control the multi-agent system. One remaining issue is that the parameters $\theta$ are optimized for using the

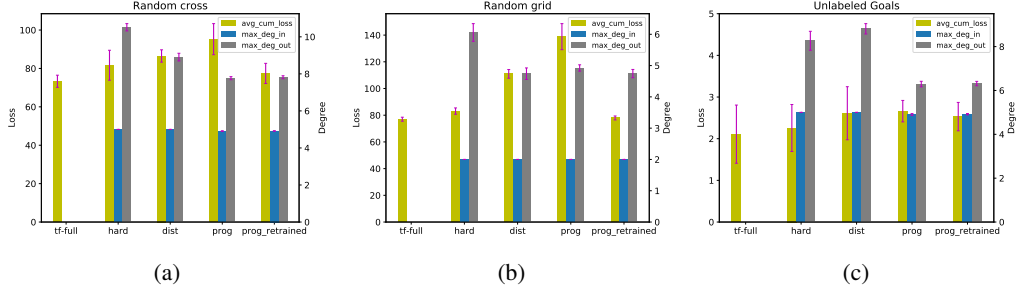

Figure 2: Statistics of cumulative loss and communication graph degrees across baselines, for (a) `random-cross`, (b) `random-grid`, and (c) `unlabeled-goals`. We omit communication degrees for `tf-full`, since it requires communication between all pairs of agents.

original soft attention weights $\alpha^{j \to i}$ rather than the hard attention weights $\alpha^{P,j \to i}$. Thus, we re-train the parameters of the transformer models in $\pi_{P,\theta}$. This training is identical to how $\pi_\theta$ was originally trained, except we use $\alpha^{P,j \to i}$ instead of $\alpha^{j \to i}$ to compute the action at each step.

## 4 Experiments

**Formation task.** We consider multi-agent formation flying tasks in 2D space [12]. Each agent has a starting position and an assigned goal position. The task is to learn a decentralized policy for the agents to reach the goals while avoiding collisions. The agents are arranged into a small number of groups (between 1 and 4): starting positions for agents within a group are close together, as are goal positions. Each agent's state $s^i$ contains its current position $x^i$ and goal position $g^i$. The observations $o^{i,j} = x^j - x^i + \epsilon^{i,j}$ are the relative positions of the other agents, corrupted by i.i.d. Gaussian noise $\epsilon^{i,j} \sim \mathcal{N}(0, \sigma^2)$. The actions $a^i$ are agent velocities, subject to $\|a^i\|_2 \le v_{\max}$. The reward at each step is $r(s,a) = r^g(s,a) - r^c(s,a)$, where the goal reward $r^g(s,a) = -\sum_{i \in [N]} \|x^i - g^i\|_2$ is the negative sum of distances of all agents to their goals, and the collision penalty $r^c(s,a) = \sum_{i,j \in [N], i \ne j} \max\{p_c(2 - \|x^i - x^j\|_2/d_c), 0\}$ is the hinge loss between each pair of agents, where $p_c$ is the collision penalty weight and $d_c$ is the collision distance.

We consider two instances. First, `random-cross` contains up to 4 groups; each possible group occurs independently with probability 0.33. The starting positions in each group (if present) are sampled uniformly randomly inside 4 boxes with center $b$ equal to $(-\ell, 0)$, $(0, -\ell)$, $(\ell, 0)$, and $(0, \ell)$, respectively, and the goal positions of each group are sampled randomly from boxes with centers at $-b$. The challenge is that agents in one group must communicate with agents in other groups to adaptively choose the most efficient path to their goals. Second, `random-grid` contains 3 groups with starting positions sampled in boxes centered at $(-\ell, 0)$, $(0, 0)$, and $(\ell, 0)$, respectively, and the goal positions are sampled in boxes centered at randomly chosen positions $(b_x, b_y) \in \{-\ell, 0, \ell\}^2$ (i.e., on a $3 \times 3$ grid), with the constraint that the starting box and goal box of a group are adjacent and the boxes are all distinct. The challenge is that each agent must learn whom to communicate with depending on its goal.

**Unlabeled goals task.** This task is a cooperative navigation task with unlabeled goals [18] that has $N$ agents along with $N$ goals at positions $g_1, ..., g_N$ (see Figure 5 in the appendix). The task is to drive the agents to cover as many goals as possible. We note that this task is not just a navigation task. Since the agents are not pre-assigned to goals, there is a combinatorial aspect where they must communicate to assign themselves to different goals. The agent state $s^i$ is its own position $x^i$ and the positions of the goals (ordered by distance at the initial time step). The observations $o^{i,j}$ are the relative positions to the other agents, corrupted by Gaussian noise. The actions $a^i = (p_1^i, \cdots, p_l^i, \cdots, p_N^i)$ are the weights (normalized to 1) over the goals; the agent moves in the direction of the weighted sum of goals—i.e., its velocity is $a^i = \sum_{k \in [N]} p_k^i (g_k - x^i)$. The reward is $r(s,a) = \sum_{k \in [N]} \max_{i \in [N]} p_k^i - N$—i.e., the sum over goals of the maximum weight that any agent assigns to that goal minus $N$.

**Baselines.** We consider four baselines. (i) Fixed communication (`dist`): A transformer, but where other agents are masked based on distance so each agent can only attend to its $k$ nearest neighbors. We find this model outperforms GCNs with the same communication structure [12], since its attention

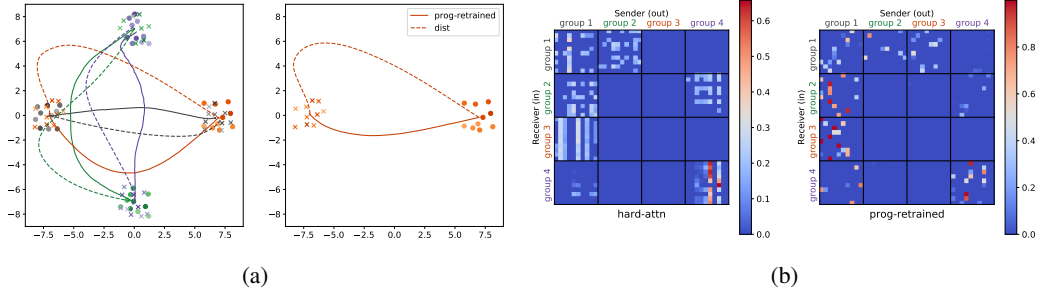

(a)                                                    (b)

Figure 3: (a) For `random-cross`, trajectories taken by each group (i.e., averaged over all agents in that group) when all four groups are present (left) and only one group is present (right), by `prog-retrained` (solid) and `dist` (dashed). Initial positions are circles and goal positions are crosses. (b) Attention weights for `hard-attn` and `prog-retrained` at a single step near the start of a rollout, computed by the agent along the $y$-axis for the message from the agent along the $x$-axis.

parameters enable each agent to re-weight the messages it receives. (ii) Transformer (`tf-full`): The oracle transformer policy from Section 3.1; here, each agent communicates with all other agents. (iii) Transformer + hard attention (`hard-attn`): The transformer policy, but where the communication degree is reduced by constraining each agent to only receive messages from the $k$ other agents with the largest attention scores. Note that this approach only minimizes the maximum in-degree, not necessarily the maximum out-degree; minimizing both would require a centralized algorithm. (iv) Transformer + program (`prog`): An ablation of our approach that does not retrain the transformer after synthesizing the programmatic communication policy (i.e., it skips the step in Section 3.5). (v) Transformer + retrained program (`prog-retrain`): Our full approach.

The tasks `random-cross` and `random-grid` perform 1 round of communications per time step for all the baselines, while the `unlabeled-goals` task uses 2 rounds of communications. For all approaches, we train the model with 10k rollouts. For synthesizing the programmatic policy, we build a dataset using 300 rollouts and run MCMC for 10000 steps. We retrain the transformer with 1000 rollouts. We constrain the maximum in-degree to be a constant $d_0$ across all approaches (except `tf-full`, where each agent communicates with every other agent); for `dist` and `hard-attn`, we do so by setting the communication neighbors to be $k = d_0$, and for `prog` and `prog-retrain`, we choose the number of rules to be $K = d_0$. This choice ensures fair comparison across approaches.

**Results.** We measure performance using both the loss (i.e., negative reward) and maximum communication degree (i.e., maximum degree of the communication graph), averaged over the time horizon. Because the in-degree of every agent is constant, the maximum degree equals the in-degree plus the maximum out-degree. Thus, we report the maximum in-degree and the maximum out-degree separately. Results are in Figure 2; we report mean and standard deviation over 20 random seeds.

For `random-cross` and `random-grid` tasks, our approach achieves loss similar to the best loss (i.e., that achieved by the full transformer), while simultaneously achieving the best communication graph degree. In general, approaches that learn communication structure (i.e., `tf-full`, `hard-attn`, and `prog-retrained`) perform better than having a fixed communication structure (i.e., `dist`). In addition, using the programmatic communication policy is more effective at reducing the maximum degree (in particular, the maximum out-degree) compared with thresholding the transformer attention (i.e., `hard-attn`). Finally, retraining the transformer is necessary for the programmatic communication policy to perform well in terms of loss. For `unlabeled-goals` task, our approach performs almost similar to `dist` baseline and slightly worse than `tf-full` baseline, but achieves a smaller maximum degree. Moreover, the loss is significantly lower than the loss of 4.13 achieved when no communications are allowed. We give additional results and experimental details in the appendix.

**Learning whom to communicate with.** Figure 3a shows two examples from `random-cross`: all four groups are present (left), and only a single group is present (right). In the former case, the groups must traverse complex trajectories to avoid collisions, whereas in the latter case, the single group can move directly to the goal. However, with a fixed communication structure, the policy `dist` cannot decide whether to use the complex trajectory or the direct trajectory, since it cannot communicate

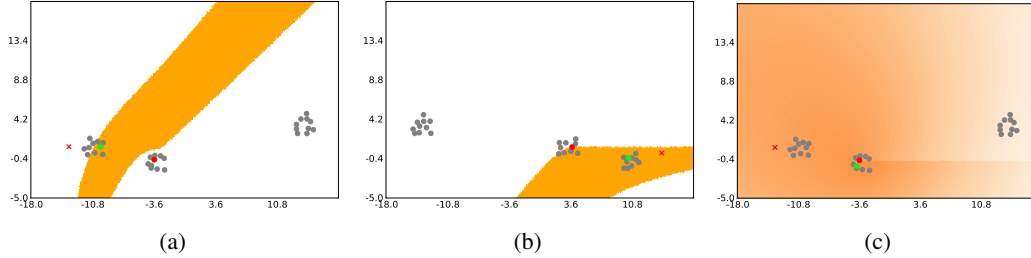

Figure 4: Visualization of a programmatic communication policy for `random-grid`, which has two rules—one determinitsic and one nondeterministic. The red circle denotes the agent making the decision, the red cross denotes its goal, and the green circle denotes the agent selected by the rule. (a,b) Visualization of the nondeterministic rule for two configurations with different goals; orange denotes the region where the filter condition is satisfied (i.e., the rule chooses a random agent in this region). (c) Visualization of the deterministic rule, showing the score output by the map operator; darker values are higher (i.e., the rule chooses the agent with the darkest value).

with agents in other groups to determine if it should avoid them. Thus, it always takes the complex trajectory. In contrast, our approach successfully decides between the complex and direct trajectories.

**Reducing the communication degree.** Figure 3b shows the attention maps of `hard-attn` and `prog-retrained` for the `random-cross` task at a single step. Agents using `hard-attn` often attend to messages from a small subset of agents; thus, even if the maximum in-degree is low, the maximum out-degree is high—i.e., there are a few agents that must send messages to many other agents. In contrast, `prog-retrained` uses randomness to distribute communication across agents.

**Understanding the learned program policy.** Figure 4 visualizes a programmatic communication policy for `random-grid`. Here, (a) and (b) visualize the nondeterministic rule for two different configurations. As can be seen, the region from which the rule chooses an agent (depicted in orange) is in the direction of the goal of the agent, presumably to perform longer-term path planning. The deterministic rule (Figure 4c) prioritizes choosing a nearby agent, presumably to avoid collisions. Thus, the rules focus on communication with other agents relevant to planning.

## 5 Conclusion

We have proposed an approach for synthesizing programmatic communication policies for decentralized control of multi-agent systems. Our approach performs as well as state-of-the-art transformer policies while significantly reducing the amount of communication required to achieve complex multi-agent planning goals. We leave much room for future work—e.g., exploring other measures of the amount of communication, better understanding what information is being communicated, and handling environments with more complex observations such as camera images or LIDAR scans.

## Broader Impact

Broadly speaking, reinforcement learning has the promise to significantly improve the usability of robotics in open-world settings. Our work focuses on leveraging reinforcement learning to help solve complex decentralized multi-agent planning problems, specifically by helping automate the design of communication policies that account for computational and bandwidth constraints. Solutions to these problems have a wide range of applications, both ones with positive societal impact—e.g., search and rescue, disaster response, transportation, agriculture, and constructions—and ones with controversial or negative impact—e.g., surveillance and military. These applications are broadly true of any work that improves the capabilities of multi-agent systems such as self-driving cars or drones. Restricting the capabilities of these systems based on ethical considerations is a key direction for future work.

Beyond communication constraints, security and robustness are important requirements for multi-agent systems. While we do not explicitly study these properties, a key advantage of reduced communication is to improve the resilience and robustness of the system and reduce the probability of failure, since there are fewer points of failure. Furthermore, communication policies that include

stochastic rules are typically more robust since they can replace a broken communication link with another randomly selected link without sacrificing performance.

Furthermore, our research may have applications in other areas of machine learning. In general, there has been growing interest in learning programmatic representations to augment neural network models to improve interpretability, robustness, and generalizability. Along these dimensions, our work could potentially impact other applications such as NLP where transformers are state-of-the-art. In particular, our work takes a step in this direction by replacing soft attention weights in transformers with programmatic attention rules. The programmatic nature of these weights makes them much easier to interpret, as does the fact that the weights are hard rather than soft (since we now have a guarantee that parts of the input are irrelevant to the computation).

## Acknowledgments and Disclosure of Funding

We gratefully acknowledge support from ONR Grant N00014-17-1-2699, IBM Research, DARPA Grant HR001120C0015, Boeing, NSF Grant 1917852, ARL Grant DCIST CRA W911NF-17-2-0181, NSF Grants CNS-1521617 and CCF-1910769, ARO Grant W911NF-13-1-0350, ONR Grant N00014-20-1-2822, ONR Grant N00014-20-S-B001, DARPA Grant FA8750-19-2-0201, ARO Grant W911NF-20-1-0080, IBM Research, and Qualcomm Research. The views expressed are those of the authors and do not reflect the official policy or position of the Department of Defense or the U.S. Government.

## Footnotes

[2]The code and a video illustrating the different tasks are available at `https://github.com/jinala/multi-agent-neurosym-transformers`.

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
