[Supplementary Material]

## A  Extending to Multi-Round Communications

The formulation in Section 3 can be extended to multiple rounds of communications per time step. For the transformer architecture with two rounds of communications, first there is an *internal network* $\pi_\theta^H : \mathbb{R}^{d_{\text{in}}} \to \mathbb{R}^{d_{\text{out}}}$ that combines the state and the cumulative message into an internal vector $h^i = \pi_\theta^H \left( s^i, \ \sum_{j=1}^N \alpha^{j \to i} m^{j \to i} \right)$. Next, we compute the next round of messages as $m'^{i \to j} = \pi_\theta^{M'}(h^i, o^{i,j})$ which replaces the state $s^i$ in the original equation with the internal state $h^i$. New keys and queries are also generated as $k'^{i,j} = \pi_\theta^{K'}(s^i, o^{i,j})$ and $q'^i = \pi_\theta^{Q'}(s^i)$, but these still use the original state $s^i$. Finally, the Equations 2 and 1 are repeated to compute the action. This architecture can be extended similarly to an arbitrary number of rounds of communications. A programmatic policy for $R$ rounds of communications will have $R$ different programs (one for each round). We synthesize these programs independently. To synthesize the communication program $P_r$ for the $r$-th round of communication, we use the hard attention weights $\alpha^{P_r}$ for the $r$-th round and use the original soft attention weights for the other rounds $r' \neq r$ to compute the synthesis objective $\tilde{J}(P_r)$.

## B  Experimental Details

The code and a short video illustrating the different tasks used in the paper can be found in `https://github.com/jinala/multi-agent-neurosym-transformers`. Figure 5 shows the initial and goal positions for the unlabeled goals task, along with attention maps produced by our program policies for the two rounds of communications at a particular timestep.

There are four main hyper-parameters in our synthesis algorithm.

- $\tilde{\lambda}$ in Section 3.4: This parameter strikes a balance between minimizing the difference in the actions (with and without hard attention) and minimizing the maximum communication degree. We use $\tilde{\lambda} = \{0.3, 0.5, 0.7, 1.0\}$.

- The number of rules in the program $= \{2, 3, 4, 5\}$.

- The depth of the Boolean conditions in the filter expressions $= 2$.

- The feature map $\phi$ used in the filter predicates and the map functions. We have 2 versions: 1) for every vector $(x, y)$ in the state $s$ and the observations $o$, we also encode the norm $\sqrt{x^2 + y^2}$ and the angle $\tan^{-1}(y/x)$ as part of the features; 2) on top of 1, we add quadratic features $(x_s x_o, x_s y_o, y_s x_o, y_s y_o)$ where $(x_s, y_s)$ is the state and $(x_o, y_o)$ is the observation.

We used cross validation to choose these parameters. In particular, we chose the ones that produced the lowest cumulative reward on a validation set of rollouts; if the cumulative rewards are similar, we chose the ones that reduced the communication degree.

$\quad\quad$ (a) $\quad\quad\quad\quad\quad\quad$ (b) $\quad\quad\quad\quad\quad\quad$ (c)

Figure 5: Unlabeled goals task: (a) Initial positions of the agents and the locations of the goals to cover (b) Final configuration of the agents where 8 out of the 10 goals are covered (c) Attention maps of `prog-retrained` for the two rounds of communication.

Figure 6: Statistics of cumulative loss and communication graph degrees for the additional baselines, for (a) `random-cross`, (b) `random-grid`, and (c) `unlabeled-goals`.

Figure 7: Comparing program policy with RL policy that treats communications as actions. RL1 and RL2 correspond to two different hyper-parameters in the policy gradient algorithm.

## C  Additional Baselines

We compare to two additional baselines: (i) an ablation of our approach that learns only deterministic rules—i.e., rules with `random` are excluded from the search space (`det-prog` and `det-prog-retrained`), and (ii) a learned communication policy in the form of a decision tree (`dt` and `dt-retrained`). For (ii), to train the decision tree, we constructed a supervised dataset by (i) collecting the soft-attentions from the transformer model, and (ii) solving the global hard-attention problem at each timestep to ensure that the maximum degree (both in-degree and out-degree) is at most $k$, where $k$ is chosen as described in Section 4 (i.e., to match the number of rules in our programmatic communication structure). Then, we train the decision tree using supervised data on this dataset.

Figure 6 shows the performance using both the loss and the maximum communication degree for these two baselines. The decision tree baselines (`dt` and `dt-retrained`) perform poorly in-terms of the communication degree for all the tasks, demonstrating that domain-specific programs that operate over lists are necessary for the communication policy to reduce communication.

The deterministic baseline (`det-prog-retrained`) achieves a similar loss as `prog-retrained` for the `random-cross` and `random-grid` tasks; however, it has worse out-degrees of communication. For these tasks, it is most likely difficult for a deterministic program to distinguish the different agents in a group; thus, all agents are requesting messages from a small set of agents. For the `unlabeled goals` task, the deterministic baseline has a lower degree of communication but has higher loss than `prog-retrained`. Again, we hypothesize that the deterministic rules are insufficient for an agent to distinguish the other agents, which led to a low in-degree (and consequently low out-degree), which is not sufficient to solve the task.

## D  Comparison to Communication Decisions as Actions

The multi-agent communication problem can be formulated as an MDP where decisions about which agents to communicate with are part of the action. We performed additional experiments to compare to this approach. Since the action space now includes discrete actions, we use the policy gradient

Figure 8: Random grid task with noisy communications.

algorithm to train the policy. We tuned several hyper-parameters including (i) weights for balancing the reward term with the communication cost, (ii) whether to use a shaped reward function, and (iii) whether to initialize the policy with the pre-trained transformer policy.

Results are shown in Figure 7. Here, rl1 is the baseline policy that achieves the lowest loss across all hyper-parameters we tried; however, this policy has a very high communication degree. In addition, rl2 is the policy with lowest communication degree; however, this policy has very high loss.

As can be seen, our approach performs significantly better than the baseline. We believe this is due to the combinatorial blowup in the action space—i.e., there is a binary communication decision for each pair of agents, so the number of communication actions is $2^{N-1}$ per agent and $2^{N(N-1)}$ for all agents (where $N$ is the number of agents). Our approach addresses this challenge by using the transformer as a teacher.

# E  Case Study with Noisy Communication

We consider a new benchmark based on the random grid task, but where the communication link between any pair of agents has a 50% probability of failing. The results are shown in Figure 8. As can be seen, the programmatic communication policy has similar loss as the transformer policy while simultaneously achieving lower communication degree. Here, the best performing policy has four rules (i.e., $K = 4$), whereas for the previous random grid task, the programmatic policy only has 2 rules. Intuitively, each agent is attempting to communicate with more of the other agents to compensate for the missing communications.