[Reviews · NeurIPS 2020]

Review 1

Summary and Contributions: This paper studies the problem of learning multi-agent communication structures. It specifically focuses on reducing the communication overhead by constraining the maximum degree of each agents (# of other agents it can communicate with) while keeping the overall performance similar. The authors train a transformer based communication policy which is fully connected and refines the policy using a simple rule. After the communication graph is sparsified, the transformer base policy is retrained to compute actions. Overall, the paper makes the following contributions: - Hardening of the communication graph for reducing communication cost - Promising empirical results on a controlled multi-agent setting

Strengths: - The hardening of the transformer policy outputs for sparsifying communication graph. - Retraining the transformer policy with hard communication graph to adapt the policy to the new graph

Weaknesses: - The paper focuses on sampling from the output distribution of the transformer policy for reducing communication cost but this is itself a RL problem where which agent to communicate with is the action. Instead, the paper picks a communication graph using an ad hoc rule and retrains the transformer with the hard graph as the transformer is not trained with its own predictions. This is not discussed in the paper. - Manually picked rules are very ad hoc and specific to the problem. I think claiming this a program is also misleading as there is only two rules (both of which are used with very simple positivity and argmax constraints w.r.t. a metric.

Correctness: Claims and empirical methodology are somewhat correct. I am skeptical of problem formulation and program synthesis terminology.

Clarity: Paper is well written with some grammatical mistakes. See additional feedbacks.

Relation to Prior Work: It is clearly discussed.

Reproducibility: Yes

Additional Feedback: In line 265, it refers to the loss as the negative reward but for unlabeled goals task, the reward is always positive (sum of max of probabilities). Please clarify Figure-2 accordingly. In line 233, element-of is repeated. In Figure 4 (c), Visualization is misspelled. In line 273, that --> than.


Review 2

Summary and Contributions: Presents and evaluates an approach for inferring communication structures in multiagent systems. Contributes to the emerging literature on learning about communication and cooperation.

Strengths: Well written, solid experimental results, interesting (and important) problem.

Weaknesses: None obvious to me.

Correctness: Everything seems in order as far as I can see.

Clarity: Pretty good.

Relation to Prior Work: Yes, insofar as I can see.

Reproducibility: Yes

Additional Feedback: Overall solid piece of work


Review 3

Summary and Contributions: The paper introduces a new algorithm for routing communication in collaborative MARL. The key idea is first to use a transformer NN to infer the communication pattern and then discretise it using MCMC. The resulting communication policies are compact - less messaging - and perform well as shown in experiments.

Strengths: The paper is clear and makes a good use of examples. It addresses a relevant problem and the proposed method is new and interesting. The work is well positioned within the literature.

Weaknesses: 1. Although the proposed method works well on the example domain, it's not shown that the method would generalise beyond that (neither theoretically nor empirically) 2. Just one domain for the experiments limits how convincing the experimental results are. It would instructive to see how the method works in domains where: a) full communication between all agents is required (does it still work and discover that?) b) communication is noisy (e.g. sometimes the message fails to arrive) c) simply a different domain, which is not navigation - capture the flag, for example 3. Novelty is somewhat limited. It is novel and interesting to use transformer weights to build a skeleton for a combinatorial optimisation, but it isn't clear that this is generally applicable to other problems than collaborative navigation.

Correctness: Yes

Clarity: Yes

Relation to Prior Work: YEs

Reproducibility: Yes

Additional Feedback:


Review 4

Summary and Contributions: The authors introduce a novel method for generating an efficient communication graph between different agents, based on first training a Transformer architecture to generate actions and messages for each agent, second conducting a combinatorial search over communication graphs, guided by the Transformer policy to produce hard attention weights for messages, and finally retraining the Transformer with these hard attention weights.

Strengths: - So far as I am aware, the methods introduced by this paper are novel. The training paradigm is particularly interesting, since the Transformer-guided program synthesis objective has potentially wide applicability. - The algorithm is clearly and concisely explained, and would be reproducible from the description, in my opinion. - Anecdotally, the programs learned by the system are reasonable in the environments considered.

Weaknesses: - The method relies on each agent having observations of other agents (o^{i,j}). This seems like a very strong assumption, given that the motivation for this work was to lower the communication bandwidth necessary. The authors should comment on how this requirement could be weakened to allow scaling to more complex environments. - The results are not presented quite clearly enough. The "loss" in Figure 2 is not clearly defined, and it would be much clearer to use "reward" as the y-axis in these Figures. The overlapping error bars in many of the results call into question the significance of the findings. Perhaps the authors can comment on these, or strengthen their method to reduce the performance variance? - There are several unclear points in the text. In line 154, what does \pi^C look like mathematically? In line 207, what is the evidence that the benefits outweigh the costs (I believe I can read this off from the ablation study, but it would be nice to point this out explicitly)? In line 220, why is Gaussian noise added?

Correctness: empty

Clarity: empty

Relation to Prior Work: empty

Reproducibility: Yes

Additional Feedback: Response to authors: I am satisfied that your responses to my comments address the concerns. Please do include these clarifications in any future version of the paper. I have therefore raised my score by 1 point.

[Author Response · NeurIPS 2020]

We thank the reviewers for their comments and will do our best to address them in our paper.

**R1, comparison to RL with communications as actions.** We agree that the multi-agent communication problem can be formulated as an MDP where decisions about which agents to communicate with is part of the action. We performed additional experiments to compare to this approach. As seen in the plots below, this approach performs poorly compared to ours. We believe this is because of the combinatorial blowup in the action space—i.e., there is a binary communication decision for each pair of agents, so the number of communication actions is $2^{N(N-1)}$ (where $N$ is the number of agents).

Since the action space now includes discrete actions, we use the policy gradient algorithm to train the policy. We tuned several hyperparameters including (i) weights for balancing the reward term with the communication cost, (ii) whether to use a shaped reward function, and (iii)

whether to initialize the policy with the pretrained transformer policy. In the figure, rl1 corresponds to the baseline policy that achieves the lowest loss across all hyperparameters we tried; however this policy has a very high communication degree. In addition, rl2 corresponds to the policy with lowest communication degree, but this policy has very high loss. In summary, learning the communication policy directly using RL is challenging due to the large discrete action space, and our proposed approach addresses this challenge by using the transformer as a teacher.

**R1, "manually picked" rules.** The communication rules used to solve each benchmark are **not chosen manually**. Instead, they are automatically learned by the search algorithm (based on MCMC) in Section 3.4. The search space is encoded by the domain-specific language (DSL) in Section 3.2. In particular, this search space consists of programs composed of arbitrary combinations of deterministic rules (with the argmax operator) and stochastic rules (with the random operator). In addition, the search algorithm must determine (i) what predicate $B$ to use in each filter operator, (ii) what function $F$ to use in each map operator, and (iii) the weights $\beta$ in each predicate/function. Finally, programs can contain more than two rules. The number of rules is a hyperparameter; we choose this parameter using cross-validation.

**R1: program synthesis terminology.** We uses the term "program" since the rules rely on map/filter operators that cannot be captured by if-then-else rules. Also, our DSL is similar to the ones used in prior work—e.g.:

Verma, A. et al. Programmatically interpretable reinforcement learning. ICML, 2018.

**R1, Figure 2.** In Figure 2c in our paper, the loss is computed as $(10 - \text{reward})$. We will clarify this in our paper.

**R3, full communication.** In our approach, the number of rules in the program is a hyperparameter $K$. Given $K$, our algorithm learns a program that optimizes performance while communicating with at most $K$ other agents (in terms of in-degree). In our experiments, the range for this hyperparameter is $K \in \{2, 3, 4, 5\}$. When we set this hyperparameter to be large, we empirically observe that each agent communicates with many other agents, so we expect that it would learn the full-communication policy as long as $K$ is sufficiently large. Our algorithm uses cross-validation to automatically choose an appropriate $K$ in the given range of possibilities.

**R3, noisy communication.** We have added a new benchmark based on the existing random grid task, but where the communications between any two pairs of agents has a 50% probability of failing. The results are shown in the adjacent figure. As can be seen, the programmatic communication policy has similar loss as the transformer policy while simultaneously achieving lower communication degree. Here, the best performing policy has four rules (i.e., hyperparameter $K = 4$), whereas for the existing random grid task, the programmatic policy has 2 rules. Intuitively, agents are attempting to communicate with more of the other agents to compensate for lost communications.

**R3, non-navigation domains.** We note that the unlabeled goals task in the paper is not just a navigation task. In this task, the agents are not pre-assigned goals, so there is a combinatorial aspect where they must communicate to assign themselves to different goals. The main difference between this task and capture the flag is that the goals are static, whereas they are moving (typically adversarially) in capture the flag. We believe our approach can be applied even in adversarial settings, and will do our best to add an additional task along these lines to our paper.

[Meta-Review · NeurIPS 2020]

The paper proposes an approach for inferring the communication graph in multi-agent systems. It combines a gradient-based optimization with a discretization or “hardening” step. The method addresses a relevant problem, is reasonably well explained, and produces promising empirical results. In their initial reviews the reviewers expressed a number of concerns, these were, however, addressed at least in parts by the author response, and ultimately all reviewers recommend acceptance. One remaining caveat is the experimental evaluation which could be strengthened, e.g. by demonstrating that the approach works across a broader range of problems. Furthermore, the authors are strongly encouraged to incorporate the clarifications provided to the reviewers as part of the author response.